# Learning from Demonstrations with Energy based Generative Adversarial Imitation Learning

## Abstract

Traditional reinforcement learning methods usually deal with the tasks with explicit reward signals. However, for vast majority of cases, the environment wouldn't feedback a reward signal immediately. It turns out to be a bottleneck for modern reinforcement learning approaches to be applied into more realistic scenarios. Recently, inverse reinforcement learning (IRL) has made great progress in making full use of the expert demonstrations to recover the reward signal for reinforcement learning. And generative adversarial imitation learning is one promising approach. In this paper, we propose a new architecture for training generative adversarial imitation learning which is so called energy based generative adversarial imitation learning (EB-GAIL). It views the discriminator as an energy function that attributes low energies to the regions near the expert demonstrations and high energies to other regions. Therefore, a generator can be seen as a reinforcement learning procedure to sample trajectories with minimal energies (cost), while the discriminator is trained to assign high energies to these generated trajectories. In detail, EB-GAIL uses an auto-encoder architecture in place of the discriminator, with the energy being the reconstruction error. Theoretical analysis shows our EB-GAIL could match the occupancy measure with expert policy during the training process. Meanwhile, the experiments depict that EB-GAIL outperforms other SoTA methods while the training process for EB-GAIL can be more stable.

## 1 Introduction

Motivated by applying reinforcement learning algorithms into more realistic tasks, we find that most realistic environments cannot feed an explicit reward signal back to the agent immediately. It becomes a bottleneck for traditional reinforcement learning methods to be applied into more realistic scenarios. So how to infer the latent reward function from expert demonstrations is of great significance. Recently, a lot of great work have been proposed to solve this problem. They are also successfully applied in scientific inquiries, such as Stanford autonomous helicopter Abbeel et al. (2006) Abbeel et al. (2007) Ng et al. (2004) Coates et al. (2008) Abbeel et al. (2008a) Abbeel et al. (2010), as well as practical challenges such as navigation Ratliff et al. (2006) Abbeel et al. (2008b) Ziebart et al. (2008) Ziebart et al. (2010) and intelligent building controls Barrett & Linder (2015).

The goal of imitation learning is to mimic the expert behavior from expert demonstrations without access to a reinforcement signal from the environment. The algorithms in this field can be divided into two board categories: behavioral cloning and inverse reinforcement learning. Behavioral cloning formulate this problem as a supervised learning problem which aims at mapping state action pairs from expert trajectories to policy. These methods suffer from the problem of compounding errors (covariate shift) which only learn the actions of the expert but not reason about what the expert is trying to achieve. By the contrast, inverse reinforcement learning recovers the reward function from expert demonstrations and then optimize the policy under such an unknown reward function.

In this paper, we propose energy-based generative adversarial imitation learning which views the discriminator as an energy function without explicit probabilistic interpretation. The energy function computed by the discriminator can be viewed as a trainable cost function for the generator, while the discriminator is trained to assign low energy values to the regions of expert demonstrations, and

higher energy values outside these regions. We use an auto-encoder to represent the discriminator, and the reconstruction error is thought to be the energy. There are many other choices to learn the energy function, but an auto-encoder is quite efficient.

Our main contributions are summarized as follows.

- An EB-GAIL framework with the discriminator using an auto-encoder architecture in which the energy is the reconstruction error.
- Theoretical analysis shows that the policy rolls out the trajectories that are indistinguishable from the distribution of the expert demonstrations by matching the occupancy measure with the expert policy.
- Experiments show that EB-GAIL outperforms several SoTA imitation learning algorithms while the training process for EB-GAIL can be more stable.

## 2 BACKGROUND

In this section, we'll briefly introduce the basic concepts in direct reinforcement learning Sutton & Barto (1998) Sugiyama (2015), inverse reinforcement learning Ng & Russell (2000), imitation learning Bagnell (2015), and energy based models LeCun et al. (2006).

### 2.1 DIRECT REINFORCEMENT LEARNING

Reinforcement Learning Sutton & Barto (1998) Sugiyama (2015), which is usually used for sequential decision making problems, can help us to learn from the interactions with the environment. The process for direct reinforcement learning is that at each time step, the agent receives a state $s_t$ and chooses an action $a_t$ from the action space $A$, following a stochastic policy $\pi(a_t|s_t)$. After that, the environment transit to a new state $s_{t+1}$ and gives a scalar reward $r_t$ back to the agent. This process continues until the agent reaches a terminal state.

In this case, the training model can be thought as a Markov decision process (MDP), which is a tuple $(S, A, T, \gamma, D, R)$. In this tuple, $S$ is the state space; $A$ is the action space; $T = P_{sa}$ is a probability matrix for state transitions, owing to the environment dynamics; $\gamma \in (0, 1]$ is a discount factor; $D$ is the initial-state transition distribution; and $R : S \rightarrow A$ is the reward function, which is assumed to be bounded in absolute value by 1. In addition, we also define that a policy $\pi$ is a mapping from states to probability distributions over actions, which is also called a stochastic policy.

For a certain task, the goal of direct reinforcement learning is to maximize the total future reward. For simplification, we define the value function to be a prediction of the total future reward, which can be shown as a discounted future reward: $V = \sum_{t=0}^{\infty} \gamma^t R_t$. Besides, we also define the action value function as $Q(s, a) = E[R_t|s_t = s, a_t = a]$, which is the expected return for selecting action $a$ in state $s$. According to the Bellman Optimality, an optimal action value function $Q^*(s, a)$ is the maximum action value achievable by any policy for state $s$ and action $a$.

### 2.2 INVERSE REINFORCEMENT LEARNING

The goal of inverse reinforcement learning is to infer the reward signal with respect to the expert demonstrations which are assumed to be the observations of optimal behaviors Ng & Russell (2000).

In the past decade, a lot of great work have been proposed towards enlarging the ability of reward function representation. Take some traditional methods for example, in 2010, FIRL Levine & Popvić (2010) was proposed to learn a set of composites features based on logical conjunctions with nonlinear functions for the reward signal. Later, non-parametric methods based on Gaussian Process(GP) Rasmussen & Williams (2006) are proposed to enlarge the function space of latent reward to allow for non-linearity in Levine & Popvić (2010). For undertaking the learning of an abstract structure with smaller data sets, Jin et al. Jin & Spanos (2015) combined deep belief networks and gaussian process to optimize the existing algorithm GP-IRL.

To solve the substantial noise in the sensor measurements, some work have been applied here such as bayesian programming, probabilistic graphical model and so on. For example, in 2007, Ramachandran et al. proposed a Bayesian nonparametric approach Ramachandran & Amir (2007)

to construct the reward function features in IRL, which is so-called Bayesian IRL. Later, Choi et al. Choi & Kim (2014) extend this algorithm by defining a prior on the composite features, which are defined to be the logical conjunctions of the atomic features.

By assuming that each trajectory can be generated by multiple locally consistent reward functions, Nguyen et al. used an expectation-maximization (EM) algorithm to learn the different reward functions and the stochastic transitions between them in order to jointly improve the likelihood of the expert's demonstrated trajectories Nguyen et al. (2015). As a result, the most likely partition of a trajectory into segments that are generated from different locally consistent reward functions selected by EM can be derived. Experiments show that the algorithm outperforms the SoTA EM clustering with maximum likelihood IRL.

## 2.3 IMITATION LEARNING

Imitation learning is a study of algorithms that can mimic the experts' demonstrations or a teachers' behavior. Unlike inverse reinforcement learning, the goal of imitation learning is to obtain a policy from teachers' behaviors rather than to recover the reward function for some certain tasks.

The algorithms in imitation learning can be classified into two categories: behavioral cloning, and inverse reinforcement learning. One fatal weakness for behavioral cloning is that these methods can only learn the actions from the teacher rather than learn the motivation from teachers' behaviors. To solve this problem, inverse reinforcement learning was proposed to recover the reward function for decision making problems. By assuming that the teachers' behavior is optimal, these methods tend to recover the reward function in a Markov decision process. So when combined with direct reinforcement learning methods, inverse reinforcement learning can realize the process for mimicking the teachers' behaviors.

## 2.4 ENERGY BASED MODEL

The essence of the energy based model LeCun et al. (2006) is to build a function that maps each point of an input space to a single scalar, which is called "energy". The learning phase is a data driven process that shapes the energy surface in such a way that the desired configurations get assigned low energies, while the incorrect ones are given high energies. Supervised learning falls into this framework: for each $x$ in the training set, the energy of the pair $(x, y)$ takes low values when $y$ is the correct label and higher values for incorrect $y$'s. Similarly, when modeling $x$ alone within an unsupervised learning setting, lower energy is attributed to the data manifold. The term contrastive sample is often used to refer to a data point causing an energy pull up, such as the incorrect $y$'s in supervised learning and points from low data density regions in unsupervised learning.

Denoted that the energy function is $\varepsilon$, the connection between probability and energy can be built through Gibbs distribution:

$$P(y|x) = \frac{exp(-\beta\varepsilon(y, x))}{\int_{y \in Y} exp(-\beta\varepsilon(y, x))}, \tag{1}$$

the denominator here is the partition function which represents the total energy in the data space and $\beta$ is an arbitrary positive constant. The formulation of this connection might seem arbitrary, but other formulation can be obtained by re-defining the energy function.

# 3 ENERGY BASED GENERATIVE ADVERSARIAL IMITATION LEARNING (EB-GAIL)

## 3.1 METHODOLOGY

The output of the discriminator goes through an objective functional in order to shape the energy function, assigning low energy to the regions near the expert demonstrations and higher energy to the other regions. In this work, we use an auto-encoder to represent the discriminator and the reconstruction error of the auto-encoder is assumed to be the energy. Meanwhile, we use a margin loss function to train EB-GAIL while one loss function is to train the discriminator and the other loss function is assumed to be the reward function for the reinforcement learning procedure.

Given a positive margin $margin$, a state-action pair $\chi_E$ sampled from expert demonstrations, and a state-action pair $\chi_i$ rolled out by a trained policy, the discriminator loss function $\mathcal{L}_D$ is formally defined by:

$$\mathcal{L}_D = D(\chi_E) + [margin - D(\chi_i)]^+ \tag{2}$$

where $[\cdot]^+ = \max(0, \cdot)$.

Meanwhile, the reward function for the reinforcement learning procedure is:

$$r(\chi_i) = -D(\chi_i) \tag{3}$$

Maximizing the total reward for the reinforcement learning is similar to minimizing the second term of $\mathcal{L}_D$. In practice, we observe that the loss function can effectively avoid gradient vanishing and mode collapse problems.

### 3.2 Using Auto-Encoder

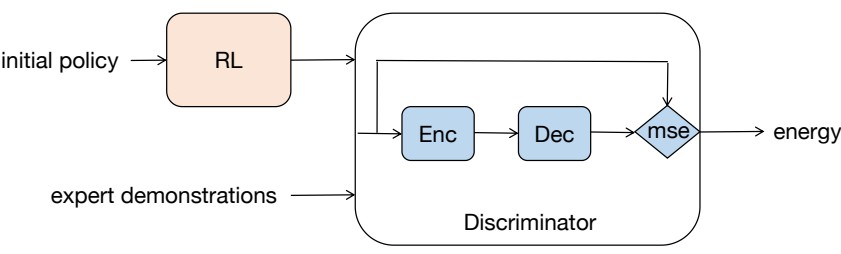

Figure 1: The architecture for EB-GAIL.

In this work, we propose that the discriminator is structured as an auto-encoder, which assigns low energy to the regions near the expert demonstrations and high energy to the other regions.

The discriminator is defined as:

$$D(x) = \|Dec(Enc(x)) - x\| \tag{4}$$

---

**Algorithm 1** Energy based Generative Adversarial Imitation Learning (EB-GAIL)

**Require:** Initial parameters of policy, discriminator $\theta_0$, $w_0$.
    Expert trajectories $\tau_E \sim \pi_E$.
    Choose a value for $margin$.
**Ensure:**
  1: **for** $i = 0$ to $N$ **do**
  2:     Sample trajectories: $\tau_i \sim \pi_{\theta_i}$ with current policy $\pi_{\theta_i}$.
  3:     Sample state-action pairs $\chi_i \sim \tau_i$ and $\chi_E \sim \pi_E$ with same batch size.
  4:     Update $w_i$ to $w_{i+1}$ by decreasing with the gradient:

$$\hat{E}_{\chi_i}[\nabla_{w_i} D_{w_i}(s,a)] + \hat{E}_{\chi_E}[\nabla_{w_i}[margin - D_{w_i}(s,a)]^+]$$

  5:     Take a policy step from $\theta_i$ to $\theta_{i+1}$, using the TRPO update rule with the reward function $-D_{w_i}(s,a)$, and the objective function for TRPO is:

$$\hat{E}_{\tau_i}[(-D_{w_i}(s,a))] - \lambda H(\pi_{\theta_i}).$$

  6: **end for**

---

Figure 1 depicts the architecture for EB-GAIL. The reinforcement learning component is trained to roll out trajectories $\tau_i$, which are the sequences of state-action pairs. The discriminator $D$ takes either expert or generated state-action pairs, and estimates the energy value accordingly. Here, we assume

the discriminator $D$ produces non-negative values. There are also many other choices for defining the energy function LeCun et al. (2006), but the auto-encoder is an efficient one.

Algorithm 1 depicts the procedure for training EB-GAIL. The first step is to sample the state-action pairs from expert demonstrations $\chi_E \sim \pi_E$. The second step is to sample the state-action pairs from the trajectories rolled out by current policy $\chi_i \sim \tau_i$ with the same batch size. Then we update the discriminator by decreasing with the gradient: $\hat{E}_{\chi_i}[\nabla_{w_i} D_{w_i}(s,a)] + \hat{E}_{\chi_E}[\nabla_{w_i}[margin - D_{w_i}(s,a)]^+]$. The fourth step is to update the policy assuming that the reward function is $-D_{w_i}(s,a)$. We runs these steps with $N$ iterations until the policy converges.

### 3.3 THEORETICAL ANALYSIS

In this section, we introduce a theoretical analysis for EB-GAIL. We show that if EB-GAIL reaches a Nash equilibrium, then the policy perfectly match the occupancy measure with the expert policy. This section is done in a non-parametric setting. We also assume that $D$ and $G$ have infinite capacity.

Firstly, we provide a definition for occupancy measure for inverse reinforcement learning.

**Definition 1.** *(occupancy measure) The agent rolls out the trajectories which can be divided into state-action pairs $(s,a)$ with policy $\pi$. This leads to the definition of occupancy measure $\rho_\pi^{s,a}(s,a)$ and $\rho_\pi^s(s)$ as the density of occurrence of states or state-action pairs:*

$$\rho_\pi^{s,a} = \sum_{t=0}^{\infty} \gamma^t P(s_t = s, a_t = s | \pi)$$

$$= \pi(a|s) \sum_{t=0}^{\infty} \gamma^t P(s_t = s|\pi) = \pi(a|s)\rho_\pi^s(s),$$

(5)

*where we assume $\gamma = 1$ for simplicity in the following paragraph.*

A basic result is that the set of valid occupancy measures $\mathcal{D} = \{\rho_\pi : \pi \in \Pi\}$ can be written as a feasible set of affine constraints: if $p_0(s)$ is the distribution of starting states and $P(s'|s,a)$ is the dynamics model, then $\mathcal{D} = \{\rho : \rho \geq 0 \text{ and } \sum_a \rho(s,a) = p_0(s) + \gamma \sum_{s',a} P(s|s',a)\rho(s',a) \quad \forall s \in \mathcal{S}\}$.

**Proposition 1.** *(Theorem 2 in Syed et al. (2008)) If $\rho \in \mathcal{D}$, then $\rho$ is the occupancy measure for $\pi_\rho(a|s) = \frac{\rho(s,a)}{\sum_{a'} \rho(s,a')}$, and $\pi_\rho$ is the only policy whose occupancy measure is $\rho$.*

Here we are justified in writing $\pi_\rho$ to denote the unique policy for an occupancy measure $\rho$. Now, let us define an IRL primitive procedure, which finds a reward function such that the expert performs better than all other policies, with the reward regularized by $\phi$:

$$\text{IRL}_\phi(\pi_E) = \arg \min_{r \in R^{S \times A}} \phi(r) + (\max_{\pi \in \Pi} H(\pi) - E_\pi[r(s,a)]) + E_{\pi_E}[r(s,a)]$$

(6)

We are interested in the policy given by running reinforcement learning procedure on the reward function which is the output of IRL.

And now we are ready to acquire the policy learned by RL under the reward function recovered by IRL:

**Proposition 2.** *(Proposition 3.2 in Ho & Ermon (2016))*

$$RL \circ IRL_\phi(\pi_E) = \arg \min_{\pi \in \Pi} -H(\pi) + \phi^*(\rho_\pi - \rho_{\pi_E})$$

(7)

*Remark* 1. Proposition 2 tells that $\phi$-regularized inverse reinforcement learning seeks a policy whose occupancy measure is close to the expert's, which is measured by the convex function $\phi^*$.

We present that the discriminator is to minimize $d_\phi(\rho_\pi, \rho_{\pi_E}) - H(\pi)$, where $d_\phi(\rho_\pi, \rho_{\pi_E}) = \phi^*(\rho_\pi - \rho_{\pi_E})$ by modifying the IRL regularizer $\phi$, so that $d_\phi(\rho_\pi, \rho_{\pi_E})$ smoothly penalizes violations in difference between the occupancy measures. And the generator is to maximize the total reward for a policy, which is $E[\sum_{t=0}^T \gamma^t r(s,a)] = E[\sum_{t=0}^T D(s,a)]$ (we assume $\gamma = 1$ for simplicity).

**Proposition 3.** *Our choice (EB-GAIL) of $\phi$ is:*

$$\phi(\rho_\pi - \rho_{\pi_E}) = \min E_{\pi_E}[D(s,a)] + E_\pi[(margin - D(s,a))^+] \tag{8}$$

The proof of this proposition is in Appendix A.1.1.

*Remark* 2. Proposition 3 tells that in our paper, the $\phi$ is determined by the loss functional, which is the margin based loss function. And we can further prove that this formulation of $\phi$ effectively help the algorithm to match the occupancy measure with expert policy during the training process.

Considered the loss functionals in section 3.1 and proposition 3, we define:

$$\begin{aligned}
V(G,D) &= \int_{\chi_i, \chi_E} \phi(\rho_\pi - \rho_{\pi_E})\rho_\pi \rho_{\pi_E} d\chi_i d\chi_E \\
&= \int_{\chi_i, \chi_E} \underbrace{\{E_{\pi_E}[D(\chi_E)] + E_\pi[(margin - D(\chi_i))^+]\}}_{\mathcal{L}_D} \rho_\pi \rho_{\pi_E} d\chi_i d\chi_E
\end{aligned} \tag{9}$$

$$U(G,D) = \int_{\chi_i} \underbrace{D(\chi_i)}_{-r(\chi_i)} \rho_\pi d\chi_i \tag{10}$$

Obviously, we train the discriminator $D$ to minimize the quantity $V$ and the generator $G$ to minimize the quantity $U$. A Nash equilibrium of EB-GAIL is a pair $(G^*, D^*)$ that satisfies:

$$\begin{aligned}
V(G^*, D^*) &\leq V(G^*, D) \qquad \forall D \\
U(G^*, D^*) &\leq U(G, D^*) \qquad \forall G
\end{aligned} \tag{11}$$

**Theorem 1.** *If the system (EB-GAIL) reaches a Nash equilibrium, then $\rho_\pi = \rho_{\pi_E}$, and $V(G^*, D^*) = m$, where $m$ represents $margin$.*

The proof of this theorem is in Appendix A.1.2.

*Remark* 3. The proof follows the idea that won't violate the equation 11 (get conclusion that $m \leq V(G^*, D^*) \leq m$). And therefore, the occupancy measure can be matched ($\rho_\pi = \rho_{\pi_E}$) when the system reaches a Nash equilibrium. Theorem 1 tells that using EB-GAIL loss functionals can help the trained policy to match the occupancy measure with the expert policy and $V(G, D)$ will converge to $margin(m)$ which indicate that the discriminator will converge while the system reaches a Nash equilibrium.

**Theorem 2.** *A Nash equilibrium of this system exists and is characterized by (1)$\rho_\pi = \rho_{\pi_E}$, and (2) there exists a constant $\gamma \in [0, m]$ such that $D^*(\chi) = \gamma$ almost everywhere, where $m$ represents $margin$.*

The proof of this theorem is in Appendix A.1.3.

*Remark* 4. Theorem 2 tells that if the Nash equilibrium of EB-GAIL exists, then it is characterized by that the trained policy could match the occupancy measure with expert policy (the trajectories rolled by the trained policy will be indistinguishable with the expert demonstrations) and $D^*(\chi) = \gamma$ almost everywhere which indicate that the discriminator could converge.

## 4 EXPERIMENTS

### 4.1 TASKS

The tasks in the presented benchmark can be divided into two categories: basic tasks and locomotion tasks. We briefly describe them in this section. We choose to implement all tasks using physics simulators rather than symbolic equations, since the former approach is less error-prone and permits easy modification of each task. Tasks with simple dynamics are implemented using Box2D, an open-source, freely available 2D physics simulator. Tasks with more complicated dynamics, such as locomotion, are implemented using MuJoCo, a 3D physics simulator with better modeling of contacts.

### 4.1.1 BASIC TASKS

We implement five basic tasks that have been widely analyzed in reinforcement learning and imitation learning literature: Cart-Pole Balancing Stephenson (1908) Donaldson (1960) Widrow (1964) Michie & Chambers (1968); Cart-Pole Swing Up Kimura & Kobayashi (1999) Doya (2000); Mountain Car Moore (1990); Acrobot Swing Up DeJong & Spong (1994) Murray & Hauser (1991) Doya (2000); and Double Inverted Pendulum Balancing Furuta et al. (1978). These relatively low dimensional tasks provide quick evaluations and comparisons of imitation learning algorithms.

### 4.1.2 LOCOMOTION TASKS

In this category, we implement seven locomotion tasks of varying dynamics and difficulty: Swimmer Purcell (1977) Coulom (2002) Levine & Koltun (2013) Schulman et al. (2015a), Hopper Murthy & Raibert (1984) Erez et al. (2011) Levine & Koltun (2013) Schulman et al. (2015a), Walker Raibert & Hodgins (1991) Erez et al. (2011) Levine & Koltun (2013) Schulman et al. (2015a), Half-Cheetah Wawrzynski (2007) Heess et al. (2015), Ant Schulman et al. (2015b), Simple Humanoid Tassa et al. (2012) Schulman et al. (2015b), and Full Humanoid Tassa et al. (2012). The goal for all these tasks is to move forward as quickly as possible. These tasks are more challenging than the basic tasks due to high degrees of freedom. In addition, a great amount of exploration is needed to learn to move forward without getting stuck at local optima. Since we penalize for excessive controls as well as falling over, during the initial stage of learning, when the robot is not yet able to move forward for a sufficient distance without falling, apparent local optima exist including staying at the origin or diving forward slowly.

## 4.2 BASELINES

In this section, we will introduce the the baseline methods used in detail.

Behavioral Cloning (BC): learning a mapping from state space to action space, with supervised learning methods. Specifically, the algorithm cannot get more information from the expert any more except the expert demonstrations. So the problem of compounding errors will occur and the performance might be poor. But in fact, the performance is quite promising.

Guided Cost Learning (GCL): the algorithm of Finn et al. (2016), which is actually a sampling based maximum entropy inverse reinforcement learning method with neural networks as the cost function. The reward function search process and policy update process are in the inner loop of the algorithm. So it will need more computation resources.

Generative Adversarial Imitation Learning (GAIL): the algorithm of Ho & Ermon (2016) which use a GAN architecture for policy improvement and reward function fitting. GAIL use the discriminator to compute the reward for some state-action pair, and then using the reward function to update the policy.

## 4.3 TRAINING SETTING

We used all the algorithms to train policies of the same neural network architecture for all tasks: two hidden layers of 100 units each, with tanh nonlinearities in between. All networks were always initialized randomly at the start of each trial. For each task, we gave BC, GCL, GAIL and our EB-GAIL exactly the same amount of environment interaction for training. We ran all algorithms 10 times over different random seeds in all environments. More information is depicted in Appendix A.2.

## 4.4 RESULTS AND DISCUSSION

Figure 2 depicts the results which are the scaled rewards for different imitation learning methods. We set the expert trajectory reward is 1.0 and random policy's reward is 0.0. In basic tasks, BC achieves comparable results compared with GCL and GAIL. And our proposed method EB-GAIL achieves expert performance on these tasks. Obviously, our EB-GAIL outperforms other SoTA methods on this evaluation metric. In locomotion tasks, GCL and GAIL achieve higher rewards than BC except for HalfCheetah and Ant. In these tasks, EB-GAIL obtains better performance than BC, GCL and

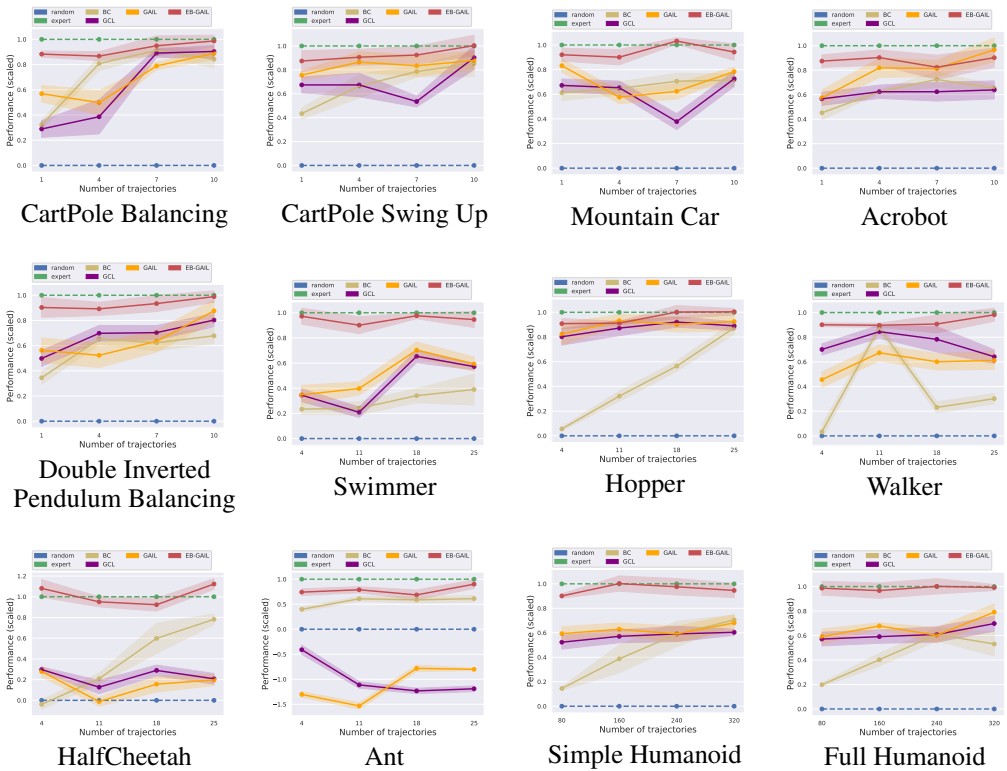

Figure 2: Comparisons for EB-GAIL and other SoTA methods on different tasks.

GAIL even with limited expert trajectories (we only use $1 \sim 10$ trajectories in basic tasks and $4 \sim 25$ trajectories in locomotion tasks except for humanoid tasks). And our proposed method EB-GAIL still achieves expert performance and outperform all the compared algorithms. These results definitely show our EB-GAIL outperforms other SoTA methods on basic tasks and MuJoCo environments. Meanwhile, the deviation for our proposed EB-GAIL is also roughly lower than other SoTA methods. It shows the stability of training our EB-GAIL.

Experiment results tells that our EB-GAIL achieves expert performance on basic tasks and locomotion tasks, and it basically outperforms other SoTA methods. Meanwhile the training process for EB-GAIL is more stable than other SoTA methods.

## 5  CONCLUSION

In this paper, we present an energy based method for generative adversarial imitation learning, which is so-called EB-GAIL. It views the discriminator as an energy function that attributes low energies to the regions near the expert demonstrations and high energies to other regions. To learn the energy of the state-action space, we use the mean square error of an auto-encoder to represent the energy function (there are still many other choices for us to represent the energy function, but the auto-encoder is quite efficient). Theoretical analysis depicts the system could match the occupancy measure with expert policy when the system reaches a Nash equilibrium. Meanwhile it also tells that if the Nash equilibrium of EB-GAIL exists, then it is characterized by that the trained policy could match the occupancy measure with expert policy and the discriminator could converge to a number between $0$ and the $margin$. Experiment results show that our EB-GAIL achieves SoTA performance, for learning a policy that can imitate and even outperform the human experts. Meanwhile, the training procedure for EB-GAIL is more stable than other SoTA methods. As we demonstrated, our method is also quite sample efficient by learning from limited expert demonstrations. We hope that our work can further be applied into more realistic scenarios.

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

## A  APPENDIX

### A.1  OMITTED PROOFS

#### A.1.1  PROOF OF PROPOSITION 3

Our choice (EB-GAIL) of $\phi$ is:

$$\phi(\rho_\pi - \rho_{\pi_E}) = \min E_{\pi_E}[D(s,a)] + E_\pi[(margin - D(s,a))^+] \tag{12}$$

*Proof.* This proof will take the JS divergence as an example. And we will prove that the cost regularizer

$$\phi_{GA}(c) = \begin{cases} E_{\pi_E}[g(c(s,a))] & if \quad c < 0 \\ +\infty & otherwise \end{cases} \tag{13}$$

where

$$g(x) = \begin{cases} -x - log(1 - e^x) & if \quad x < 0 \\ +\infty & otherwise \end{cases} \tag{14}$$

satisfies:

$$\phi_{GA}^*(\rho_\pi - \rho_{\pi_E}) = \max E_\pi[log(D(s,a))] + E_{\pi_E}[log(1 - D(s,a))]. \tag{15}$$

Using the logistic loss $log(1 + \exp(-x))$, we see that applying Proposition A.1 in Ho & Ermon (2016), we get:

$$
\begin{aligned}
\phi_{GA}^*(\rho_\pi - \rho_{\pi_E}) &= -R_\phi(\rho_\pi, \rho_{\pi_E}) \\
&= \sum_{s,a} \max_{\gamma \in R} \rho_\pi(s,a) log(\frac{1}{1 + \exp(-\gamma)}) + \rho_{\pi_E}(s,a) log(\frac{1}{1 + \exp(\gamma)}) \\
&= \sum_{s,a} \max_{\gamma \in R} \rho_\pi(s,a) log(\frac{1}{1 + \exp(-\gamma)}) + \rho_{\pi_E}(s,a) log(1 - \frac{1}{1 + \exp(-\gamma)}) \\
&= \sum_{s,a} \max_{\gamma \in R} \rho_\pi(s,a) log(\delta(\gamma)) + \rho_{\pi_E}(s,a) log(1 - \delta(\gamma)),
\end{aligned} \tag{16}
$$

where $\delta(x) = 1/(1 + \exp(-x))$ is the sigmoid function. Because the range of $\delta$ is $(0,1)$, we can write:

$$
\begin{aligned}
\phi_{GA}^*(\rho_\pi - \rho_{\pi_E}) &= \sum_{s,a} \max \rho_\pi(s,a) log d + \rho_{\pi_E}(s,a) log(1 - d) \\
&= \max \sum_{s,a} \rho_\pi(s,a) log(D(s,a)) + \rho_{\pi_E}(s,a) log(1 - D(s,a))
\end{aligned} \tag{17}
$$

which is the desired expression.

Obviously, in our EB-GAIL, we choose the expression for $\phi$ is:

$$\min E_{\pi_E}[D(s,a)] + E_\pi[(margin - D(s,a))^+] \tag{18}$$

This completes the proof. $\qquad\square$

#### A.1.2  PROOF OF THEOREM 1

**Lemma 1.** *Zhao et al. (2016) Let $a, b \geq 0$, $\phi(x) = ax + b[m - x]^+$. The minimum of $\phi$ on $[0, +\infty)$ exists and is reached in $m$ if $a < b$, and it is reached in $0$ otherwise.*

*Proof.* The function $\phi$ is defined on $[0, +\infty)$, its derivative is defined on $[0, +\infty)\backslash\{m\}$ and $\phi'(x) = a - b$ if $x \in [0, m)$ and $\phi'(x) = a$ if $x \in (m, +\infty)$.

So when $a < b$, the function is decreasing on $[0, m)$ and increasing on $(m, +\infty)$. Since it is continuous, it has a minimum in $m$.

On the other hand, if $a \geq b$ the function $\phi$ is increasing on $[0, +\infty)$, so 0 is a minimum.

This completes the proof. $\qquad\square$

**Lemma 2.** *Zhao et al. (2016) If $p$ and $q$ are probability densities, and the function $1_A(x) = 1$ if $x \in A$ otherwise $1_A(x) = 0$, then $\int_x 1_{p(x)<q(x)}dx = 0$ if and only if $\int_x 1_{p(x)\neq q(x)}dx = 0$.*

*Proof.* Let's assume that $\int_x 1_{p(x)<q(x)}dx = 0$. Then

$$
\begin{aligned}
&\int_x 1_{p(x)>q(x)}(p(x) - q(x))dx \\
&= \int_x (1 - 1_{p(x)\leq q(x)})(p(x) - q(x))dx \\
&= \int_x p(x)dx - \int_x q(x)dx + \int_x 1_{p(x)\leq q(x)}(p(x) - q(x))dx \\
&= 1 - 1 + \int_x (1_{p(x)<q(x)} + 1_{p(x)=q(x)})(p(x) - q(x))dx \\
&= \int_x 1_{p(x)<q(x)}(p(x) - q(x))dx + \\
&\quad \int_x 1_{p(x)=q(x)}(p(x) - q(x))dx \\
&= 0
\end{aligned}
\tag{19}
$$

So $\int_x 1_{p(x)>q(x)}(p(x) - q(x))dx = 0$ and since the term in the integral is always non-negative, $1_{p(x)>q(x)}(p(x) - q(x)) = 0$ for almost all $x$. And $p(x) - q(x) = 0$ implies $1_{p(x)>q(x)} = 0$, so $1_{p(x)>q(x)} = 0$ almost everywhere.

This completes the proof. $\square$

Theorem 1: If the system (EB-GAIL) reaches a Nash equilibrium, then $\rho_\pi = \rho_{\pi_E}$, and $V(G^*, D^*) = m$, where $m$ represents $margin$.

*Proof.* First we observe that

$$
\begin{aligned}
V(G^*, D^*) &= \int_{\chi_E} \rho_{\pi_E}(\chi_E)D(\chi_E)d\chi_E + \int_{\chi_i} \rho_\pi(\chi_i)[m - D(\chi_i)]^+ d\chi_i \\
&= \int_{\chi_E} (\rho_{\pi_E}(\chi_E)D(\chi_E) + \rho_{\pi^*}(\chi_E)[m - D(\chi_E)]^+)d\chi_E.
\end{aligned}
\tag{20}
$$

Lemma 1 shows: (1) $D^*(\chi) \leq m$ almost everywhere. To verify it, let us assume that there exists a set of measure non-zero such that $D^*(\chi) > m$. Let $\tilde{D}(\chi) = \min(D^*(\chi), m)$. Then $V(G^*, \tilde{D}) < V(G^*, D^*)$ which violates equation 11. (2) The function $\phi$ reaches its minimum in $m$ if $a < b$ and in $0$ otherwise. So $V(G^*, D)$ reaches its minimum when we replace $D^*(x)$ by these values. We obtain

$$
\begin{aligned}
V(G^*, D^*) &= m \int_{\chi_E} 1_{\rho_{\pi_E}(x)<\rho_{\pi^*}(\chi_E)}\rho_{\pi_E}(\chi_E)d\chi_E + \\
&\quad m \int_{\chi_E} 1_{\rho_{\pi_E}(\chi_E)\leq\rho_{\pi^*}(\chi_E)}\rho_{\pi^*}(\chi_E)d\chi_E \\
&= m \int_{\chi_E} (1_{\rho_{\pi_E}(\chi_E)<\rho_{\pi^*}(\chi_E)}\rho_{\pi_E}(\chi_E) + \\
&\quad (1 - 1_{\rho_{\pi_E}(\chi_E)<\rho_{\pi^*}(\chi_E)})\rho_{\pi^*}(\chi_E))d\chi_E \\
&= m \int_{\chi_E} \rho_{\pi^*}(\chi_E)d\chi_E + m \int_{\chi_E} 1_{\rho_{\pi_E}(\chi_E)<\rho_{\pi^*}(\chi_E)}(\rho_{\pi_E}(\chi_E) - \rho_{\pi^*})d\chi_E \\
&= m + m \int_{\chi_E} 1_{\rho_{\pi_E}(\chi_E)<\rho_{\pi^*}(\chi_E)}(\rho_{\pi_E}(\chi_E) - \rho_{\pi^*}(\chi_E))d\chi_E.
\end{aligned}
\tag{21}
$$

The second term in equation 21 is non-positive, so $V(G^*, D^*) \leq m$.

By putting the ideal generator that generates $p_{data}$ into the right side of equation 11, we get

$$\int_{\chi_E} \rho_{\pi^*}(\chi_E)D^*(\chi_E)d\chi_E \leq \int_{\chi_E} \rho_{\pi_E}(\chi_E)D^*(\chi_E)d\chi_E. \tag{22}$$

Thus by equation 20,

$$\int_{\chi_E} \rho_{\pi^*}(\chi_E)D^*(\chi_E)d\chi_E + \int_{\chi_E} \rho_{\pi^*}(\chi_E)[m - D^*(\chi_E)]^+ d\chi_E \leq V(G^*, D^*) \tag{23}$$

and since $D^*(\chi) \leq m$, we get $m \leq V(G^*, D^*)$.

Thus, $m \leq V(G^*, D^*) \leq m$, so $V(G^*, D^*) = m$. According to equation 21, we see that can only happen if $\int_\chi 1_{\rho_{\pi_E}(\chi) < \rho_\pi(\chi)} d\chi = 0$. According to Lemma2, it is true if and only if $\rho_\pi = \rho_{\pi_E}$.

This completes the proof. □

### A.1.3 PROOF OF THEOREM 2

A Nash equilibrium of this system exists and is characterized by $(1)\rho_\pi = \rho_{\pi_E}$, and (2) there exists a constant $\gamma \in [0, m]$ such that $D^*(x) = \gamma$ (almost everywhere).

*Proof.* The sufficient conditions are obvious. The necessary condition on $\pi^*$ comes from theorem 1, and the necessary condition on $D^*(\chi) \leq m$ is from the proof of theorem 1.

Let us now assume that $D^*(\chi)$ is not constant almost everywhere and find a contradiction. If it is not, then there exists a constant $C$ and a set $\mathcal{S}$ of non-zero measure such that $\forall \chi \in \mathcal{S}$, $D^*(\chi) \leq C$ and $\forall \chi \notin \mathcal{S}$, $D^*(\chi) > C$. In addition we can choose $\mathcal{S}$ such that there exists a subset $\mathcal{S}' \subset \mathcal{S}$ of non-zero measure such that $\rho_{\pi_E}(\chi) > 0$ on $\mathcal{S}'$. We can build a generator policy $\rho_0$ such that $\rho_{\pi_0}(\chi) \leq \rho_{\pi_E}(\chi)$ over $\mathcal{S}$ and $\rho_{\pi_0}(\chi) < \rho_{\pi_E}(\chi)$ over $\mathcal{S}'$. We compute

$$U(G^*, D^*) - U(G_0, D^*) = \int_\chi (\rho_{\pi_E} - \rho_{\pi_0})D^*(\chi)d\chi$$

$$= \int_\chi (\rho_{\pi_E} - \rho_{\pi_0})(D^* - C)d\chi$$

$$= \int_\mathcal{S} (\rho_{\pi_E} - \rho_{\pi_0})(D^*(\chi) - C)d\chi +$$

$$\int_{\mathcal{R}^N \setminus \mathcal{S}} (\rho_{\pi_E} - \rho_{\pi_0})(D^*(\chi) - C)d\chi > 0 \tag{24}$$

which violates equation 11.

This completes the proof. □

### A.2 EXPERIMENTS

#### A.2.1 LOCOMOTION TASKS

Figure 3 depicts locomotion tasks' environments.

#### A.2.2 TRAINING SETTING

For BC, we split a given dataset of state-action pairs into 70% training data and 30% validation data. The policy is trained with supervised learning, with minibatches of 64 examples, until validation error stops decreasing.

For GCL, we use neural network to represent the reward function, and the learning rate for policy update is 0.0001, the learning rate for reward update is 0.001, with minibatches of 64 examples.

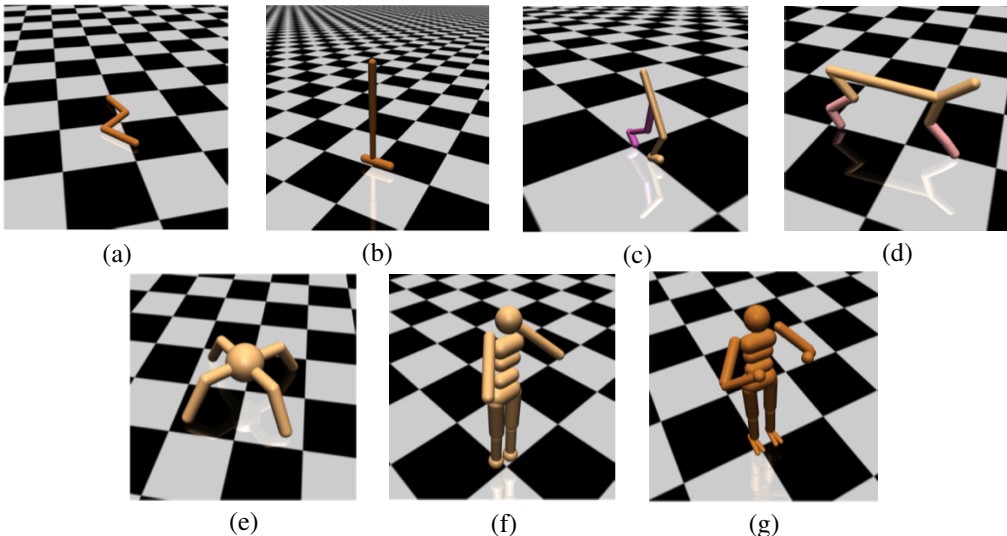

Figure 3: Illustrations for locomotion tasks: (a) Swimmer; (b) Hopper; (c) Walker; (d) Half-Cheetah; (e) Ant; (f) Simple Humanoid; and (g) Full Humanoid.

For GAIL, the discriminator networks used two hidden layers of 100 units each, with tanh nonlinearities as its architecture. And the learning rate for policy update is 0.0001, and the learning rate for discriminator update is 0.001, with minibatches of 64 examples.

For our EB-GAIL, the discriminator (auto-encoder) networks used four hidden layers of 100 units each, with tanh nonlinearities as its architecture. And the learning rate for policy update is 0.0001, the learning rate for discriminator (auto-encoder) update is 0.001, and $margin$ is 5, with minibatches of 64 examples.

