# OpenReview forum: "Learning from Demonstrations with Energy based Generative Adversarial Imitation Learning"
_ICLR.cc/2021/Conference — Reject_

### Official Review · AnonReviewer4 · 2020-10-27
**Review for Learning from Demonstrations with Energy based Generative Adversarial Imitation Learning**

**Rating:** 5
**Confidence:** 3

**Review:**

Summary
The paper proposes a new method, called EB-GAIL to deal with the Inverse Reinforcement Learning problem. EB-GAIL is an architecture for training generative adversarial imitation learning, where the discriminator is an autoencoder with the energy being the reconstruction error.

Strengths
The experiments are promising since the algorithm performs well in almost all experiments.

Weakness

The novelty of the article is unclear. Theoretical results are marginal as they are mostly from previous works.
It is hard to understand the proofs.
- What is $R_{\phi}$ in equation 16? I think that it is the same quantity as in the Generative Adversarial Imitation Learning paper.
- What is d in equation 17?
- I think there is an error in the second line of equation 21: a $\le$ is $\ge$

The paper overall is not well written. The Background section is confusing: it starts with an introduction to RL, then there is a subsection related to IRL, and at the end another subsection to imitation learning, where is another explanation of IRL. I would suggest changing this section into RL, Imitation Learning (IRL + BC).

What are the main differences between this paper and some new works as Energy-Based Imitation Learning [1] or Strictly Batch Imitation Learning by Energy-based Distribution Matching[2]?


Minor comments
- the citations are all without parentheses
- the reward is defined as a function R: S $\rightarrow$ A
- the discounted future reward has to be in expectation under the initial-state distribution, the transition model and the policy
- it is missed a citation to behavioural cloning
- it is missed a comparison with



Final comment

The document is unclear, some sections are confusing. The idea is interesting, but the paper must be edited before it can be published.

---

> ### Author Response · Authors · 2020-11-14
> **For Reviewer 4**
>
> Thanks for your review. We will dispel your concerns one by one as follows.
>
> Q1: What is Rϕ in equation 16? I think that it is the same quantity as in the Generative Adversarial Imitation Learning paper.
>
> A1: Yes.
>
> Q2: What is d in equation 17?
>
> A2: the output of the discriminator.
>
> Q3: I think there is an error in the second line of equation 21: a ≤ is a ≥?
>
> A3: Sorry, it is a typo, we will revise this in the final version of our paper.
>
> Q4: The Background section is confusing…
>
> A4: We will revise this section in the final version of our paper.
>
> Q5: What are the main differences between this paper and some new works as Energy-Based Imitation Learning or Strictly Batch Imitation Learning by Energy-based Distribution Matching?
>
> A5: Unlike EBIL and Strictly Batch Imitation Learning by Energy-based Distribution Matching , in our paper, we proposed a new method for IRL problem with theoretical guarantees. And when the generator/discriminator reaches a Nash equilibrium, the occupancy measure of the trained policy will match that of the expert policy which is the goal of IRL algorithms (reference: Umar Syed et al. Apprenticeship learning using linear programming. In International Conference on Machine Learning, 2008.). We will add some analysis in the final version of our paper. We will add more analysis and comparison with these new works as Energy-Based Imitation Learning or Strictly Batch Imitation Learning by Energy-based Distribution Matching.
>
> Q6: minor points…
>
> A6: We will revise this section in the final version of our paper.

---

### Official Review · AnonReviewer3 · 2020-10-28
**Interesting result suffering from unclear presentation**

**Rating:** 4
**Confidence:** 2

**Review:**

The authors propose a discriminator-based approach to inverse reinforcement learning (IRL). The discriminator function is trained to attain large values ("energy") on trajectories from the current policy and small values on trajectories from an expert policy. The current policy is then improved by using the negative discriminator as a reward signal. The specific discriminator suggested is an autoencoder loss. The authors continue to provide a proof that assuming their discriminator/generator attain a Nash equilibrium, the occupancy measure of the trained policy matches that of the expert policy. They follow up with demonstrating better performance of their approach compared to certain baselines when tested on a number of tasks on Physics simulators.

I believe this paper has potential. Unfortunately I don't think it is publishable in its current form, mainly due to stylistic issues and small errors collectively grave enough to distract from understanding. The language is often unclear due to the unusual grammar, bibliographic references are inserted into the text without any distinction from the surrounding text, where clearer, the language is often a bit too casual for a scientific publication, and there's other small errors that hinder understanding. For example, I believe the above description (discriminator is trained to be small/negative on expert trajectories and large on trajectories from the current policy) is correct, but it's not entirely clear from the paper as e.g. the listing in Algorithm 1 seems to indicate the opposite (comparing the formula in point 4 there with equation (2))? On reading this, I first wondered what happened for negative values of $D$, but equation (4) seems to suggest $D$ is positive at all times? This should be spelled out more properly, and $D$ should be introduced before it is first used in equation (2). I also couldn't find any description of the neural network architecture used in the experiments (the functional form of the final layer would also have cleared up my understanding of what kind of values $D$ produces).

I'm also unclear about the blanket statement "One fatal weakness for behavioral cloning is that these methods can only learn the actions from the teacher rather than learn the motivation from teachers’ behaviors." Recent breakthroughs such as "AlphaStar: Mastering the Real-Time Strategy Game StarCraft II" employing a version of behavioral cloning (specifically, "kickstarting") seem to suggest otherwise, so this statement should be given some kind of supporting reference. The same goes for the statement "These methods suffer from the problem of compounding errors (covariate shift)"; I have heard the term "covariate shift" before but I am not truely certain what it _means_.

In addition to the above, I suspect that the comparison to known baselines is not necessarily the most interesting setting to evaluate the proposed method. However, that is a minor point given the current state of this paper.

That said, I believe the proposed method is likely to be a good one and will make a worthwhile contribution to scientific progress on IRL. I would encourage the authors to invest the time to polish this submission, ideally also getting it proofread by a diverse set of colleagues. As far as I could tell, the mathematics seems correct and the underlying idea is sufficiently substantial to be published in at a ML conference once it's clarified and the presentational issues are resolved.

---

> ### Author Response · Authors · 2020-11-14
> **For Reviewer 3**
>
> Thanks for your review. We will dispel your concerns one by one as follows.
>
> Q1: The language is often unclear due to the unusual grammar, bibliographic references are inserted into the text without any distinction from the surrounding text, …
>
> A1: We will polish the paper in the final version of our paper. Thanks for your advice.
>
> Q2: couldn't find any description of the neural network architecture used in the experiments…
>
> A2: In appendix A.2.2, we describe the architecture of the neural network for the experiments. We will provide more information for the experiments in the final version of our paper.
>
> Q3: Behavioral cloning suffers from the problem of compounding errors (covariate shift), but it seems contradictory with AlphaStar.
>
> A3: Compounding errors (covariate shift) means that in each decision step, there will be a small error for predicting an action under some states, and with the steps increasing, this error will be compounded. Finally, we only learn the actions from the teacher rather than reason about what the expert wants to achieve. In AlphaStar, Behavioral cloning handles this problem by interacting with the environments. So it can achieve good performance in AlphaStar.
>
> Q4: I suspect that the comparison to known baselines is not necessarily the most interesting setting to evaluate the proposed method.
>
> A4: We will provide more baselines in our experiments as soon as possible such as AIRL. And on the other hand, our EB-GAIL is designed for better imitating the expert behaviors and the scaled reward is a good evaluation metric for testing our algorithm with other SoTA methods.

---

### Official Review · AnonReviewer1 · 2020-10-29
**More experiments are needed**

**Rating:** 5
**Confidence:** 4

**Review:**

Strong points:
1. The authors propose an auto-encoder network to replace the discriminator in the GAIL model. The authors also provide some theoretical analysis to show the proposed measure can match the occupancy measure of the expert policy.
2. The authors conduct experiments on different tasks to show the proposed method can outperform GAIL, BC and GCL.

Weak points:
1. Some key baselines are missing. There are a few methods that have been proposed and proved to outperform GAIL and GCL. Here, we just show one example. For instance,

Fu, J., Luo, K., & Levine, S. (2017). Learning robust rewards with adversarial inverse reinforcement learning. arXiv preprint arXiv:1710.11248.
These methods should be compared.

2. It is better if the authors can explain why using auto encoder as the discriminator can outperform the original GAIL.
3. The experiments are conducted under relatively small data (usually a few trajectories). Is there a reason? If sufficient data is provided, will the proposed method still perform the best? Is the proposed method specially designed for this small data scenario? The authors may want to explain these questions in more detail.

---

> ### Author Response · Authors · 2020-11-14
> **For Reviewer 1**
>
> Thanks for your review. We will dispel your concerns one by one as follows.
>
> Q1: Some key baselines are missing, such as AIRL etc.
>
> A1: We will provide more baselines in our experiments as soon as possible such as AIRL.
>
> Q2: Why using auto encoder as the discriminator can outperform the original GAIL?
>
> A2: We formulate the IRL problem as an EBM problem which assigns low energies for trajectories sampled by expert policies and high energies for other regions. And the auto encoder is one efficient choice for learning the energy function in EBMs. Under such this setting, we prove that when the generator/discriminator reaches a Nash equilibrium, the occupancy measure of the trained policy will match that of the expert policy which is the goal of IRL algorithms (reference: Umar Syed et al. Apprenticeship learning using linear programming. In International Conference on Machine Learning, 2008.).
>
> Q3: If sufficient data is provided, will the proposed method still perform the best? Is the proposed method specially designed for this small data scenario?
>
> A3: The locomotion tasks are very complex environments with large dynamics. For imitation learning algorithms, basic tasks and locomotion tasks are classical environments for testing the efficacy of these methods. The amount of data at present is enough for testing the algorithms. And if sufficient data is provided, the performances of these methods can hardly improve. We will add some analysis in the final version of our paper.

---

### Official Review · AnonReviewer2 · 2020-11-10
**Lack of novelty and rigor**

**Rating:** 4
**Confidence:** 5

**Review:**

This paper proposes a learning framework for imitation learning (IL) that uses an energy-based objective for generative adversarial imitation learning (GAIL).

Pros
+ Clarity. The paper is easy to understand and follow.
+ Theoretical analysis is sound (not novel however -- see below).
+ Benchmarking on sufficient number of environments that are standard in the literature. The baselines are however insufficient (see below).

Cons
- The major concern with this work is a severe lack of novelty. There are thousands of follow-ups to GANs. In GAIL, it was shown that GANs can be applied to imitation learning. Repeating a GAN work in the IL context consequently should offer significantly more algorithmic/empirical/theoretical insights than the current work.
- Relatedly, the paper lacks discussion and acknowledgement of related work. The most-closely related work besides GAIL is EB-GAN (Zhao et al.). This work has been cited but not discussed anywhere. Moreover, the references to this work are in the appendix. Theorem 1 essentially  follows directly from Lemma 1 and 2 (proved by Zhao et al) and Theorem 1 of the original GAN paper (Goodfellow et al.). It is unclear if there is anything novel in the analysis or a conclusion that is particularly novel to imitation learning. The subsequent analysis on occupancy measure matching also follows directly from GAIL.
- On the empirical end, the baselines are very weak in the context of the current work. Just like the literature on GANs has come a long way for improving training stability and mode coverage, the benefits of these GAN++ frameworks have also been shown in the context of imitation learning. For example, the InfoGAIL work by Li et al. shows the benefits of using the WGAN objective for IL and would have been a stronger baseline for empirical comparisons.

In summary, the work falls in short justifying the merits of EB-GAIL in the context of our existing understanding of imitation learning due to insufficient experimentation and theoretical/algorithmic analysis.

Minor
- Please use \citet and \citep appropriately for improving readability of references.

---

> ### Author Response · Authors · 2020-11-14
> **For Reviewer 2**
>
> Thanks for your review. We will dispel your concerns one by one as follows.
>
> Q1: Repeating a GAN work in the IL context consequently should offer significantly more algorithmic/empirical/theoretical insights than the current work. It is unclear if there is anything novel in the analysis or a conclusion that is particularly novel to imitation learning.
>
> A1: In this paper, we proposed a new method for IRL problem with theoretical guarantees. One contribution is that we prove by iteratively update the discriminator (energy function) and the generator (RL part), the system will reach a Nash equilibrium, and when the generator/discriminator reaches a Nash equilibrium, the occupancy measure of the trained policy will match that of the expert policy which is the goal of IRL algorithms (reference: Umar Syed et al. Apprenticeship learning using linear programming. In International Conference on Machine Learning, 2008.). We will add some analysis in the final version of our paper. Thanks for your advice.
>
>
> Q2:  the baselines are very weak in the context of the current work, for example, the InfoGAIL work by Li et al. shows the benefits of using the InfoGAN objective for imitation learning.
>
> A2: We will provide more baselines in our experiments as soon as possible such as AIRL. On the other hand, InfoGAIL is a method for inferring the latent structure of expert demonstrations in an unsupervised way. It considers the mode coverage problem in GAIL. Unlike InfoGAIL, we propose a method can better imitate expert behaviors by matching the occupancy measure with that of expert policies.
>
>
> Q3: minor ...
>
> A3: we will revise our paper carefully

---

### Public Comment · ~Jianwen_Xie1 · 2020-11-14
**It would be nice to discuss another energy-based framework for imitation learning**

Dear Authors,

I would be very nice to discuss the following paper [1], which also uses energy-based model for imitation learning.

The energy function plays the role of cost function for optimal control, and it can be learned from demonstration, such as human drivers for autonomous driving.  It uses MCMC to predict the trajectories.  The model only has one single EBM.

The EBM model foundation is in [2], which is the first paper on maximum likelihood learning of modern ConvNet-parametrized energy-based model.

The paper [1] also provides the second energy-based framework, in which an EBM is jointly trained with a generator. This framework has two components: EBM and the generator, which are more similar to your EB-GAN.  However, the EBM and the generator in [1] are not trained via adversarial training but via cooperative training [3].  Because both EBM and generator are conditional models in [1]. The conditional model actually is a fast-thinking and slow thinking framework in [4].

Related papers:

[1] Y Xu, J Xie, T Zhao, C Baker, Y Zhao, and YN Wu (2020) Energy-based continuous inverse optimal control. Machine Learning for Autonomous Driving Workshop at NeurIPS 2020.

[2] J Xie*, Y Lu*, SC Zhu, and YN Wu. A theory of generative ConvNet. International Conference on Machine Learning (ICML) 2016.

[3] J Xie, Y Lu, R Gao, SC Zhu, and YN Wu. Cooperative learning of descriptor and generator networks. IEEE Transactions on Pattern Analysis and Machine Intelligence (PAMI) 2018.

[4] Cooperative Training of Fast Thinking Initializer and Slow Thinking Solver for Multi-Modal Conditional Learning. ArXiv 2019

Thank you.

.

---

### Decision · Program_Chairs · 2021-01-07
**Final Decision**

**Decision:**

Reject

**Comment:**

This work proposes to uses an energy-based objective combined with generative adversarial networks for imitation learning. While most reviewers find the work easy to follow and come with theoretical justifications, albeit mostly followed from previous works, and good coverage of experimental results, all of them raised questions regarding the limited novelty and added contribution of the work, and missing more recent baselines. Please consider address these feedback in your future submissions.